# Fully inverse adsorption enables one-step high-purity $C_2H_2$ separation from ternary C2 mixtures in a robust porous crystal

Mingxing Zhang [1], Jingui Duan [2,3,4] ✉, Yanfei Feng[2] & Junfeng Bai[2] ✉

Direct harvesting of electronic-grade acetylene ($C_2H_2$) from ternary C2 mixtures is a great challenge due to the ubiquitous adsorption preference of conventional porous materials ($C_2H_2$ > ethylene ($C_2H_4$) > ethane ($C_2H_6$)). Here, we report a strategy to reverse this selectivity by leveraging ligand functionalization in porous crystals. Through the incorporation of trifluoromethyl/methyl groups into a pyrazole-carboxylate linker, we engineer a series of MOF-5 analogs. The optimal material, **NTU-98**, fully reverses the adsorption trend of C2 hydrocarbons ($C_2H_6$ > $C_2H_4$ > $C_2H_2$), enabling direct production of $C_2H_2$ with >99.99% purity from ternary feeds at room temperature in one-step. Combined density functional theory calculations and gas-loaded crystallographic analyses unveil the molecular mechanism: methyl groups precisely positioned within the cages enhance host-guest interactions with $C_2H_4$ and $C_2H_6$, while suppressing the binding affinity for $C_2H_2$. This work presents a porous crystal for direct $C_2H_2$ purification from ternary feeds and a blueprint for designing microporous environments targeting challenging separations.

Acetylene ($C_2H_2$) is a crucial feedstock in the chemical and semiconductor industries, where ultrahigh-purity electronic-grade $C_2H_2$ is indispensable for manufacturing carbon masks used in photolithography during advanced integrated circuit production[1]. However, industrial $C_2H_2$ streams mainly derived from cracked gas or ethane ($C_2H_6$) dehydrogenation typically contain ternary C2 hydrocarbon mixtures ($C_2H_2$, $C_2H_4$ and $C_2H_6$) with nearly identical physicochemical properties, making their separation particularly challenging[2,3]. Current purification strategies predominantly rely on solvent extraction, wherein $C_2H_2$ is selectively dissolved into organic solvents and subsequently recovered via desorption[4]. While effective, this process is constrained by the steep temperature dependence of hydrocarbon solubility, necessitating energy-intensive sub-ambient conditions (e.g., −40 °C) to maximize efficiency. Furthermore, solvent regeneration generates substantial liquid waste, and the purity of recovered $C_2H_2$ often falls short of the stringent electronic-grade thresholds in a single operational cycle. These limitations underscore an urgent demand for

innovative separation technologies that simultaneously prioritize energy.

Adsorptive separation presents a compelling alternative to conventional absorption methods, particularly for applications requiring high selectivity, energy efficiency, facile regeneration, and operational versatility. Unlike solvent-dependent absorption, adsorption leverages the selective interactions between target molecules and tailored porous adsorbents, thereby enabling precise capture of trace impurities and adaptability to environmentally sensitive processes[5,6]. Microporous materials such as zeolites and activated carbon have been considered as prominent candidates for $C_2H_2$ purification[7,8]. Yet their practical utility is hampered by inherent limitations: elevated temperature for adsorbent regeneration and/or insufficient selectivity. Particularly, the high reactivity of $C_2H_2$ at elevated temperatures may produce undesirable side reactions (e.g., polymerization or decomposition), adding new impurities or blocking the pore of the adsorbent. These challenges persist even in state-of-the-art systems,

[1]College of Chemistry and Chemical Engineering, Nantong University, Nantong, Jiangsu, China. [2]State Key Laboratory of Materials-Oriented Chemical Engineering & School of Chemistry and Molecular Engineering, Nanjing Tech University, Nanjing, China. [3]State Key Laboratory of Chemistry and Utilization of Carbon-Based Energy Resources, College of Chemistry, Xinjiang University, Urumqi, China. [4]Nanjing Tech University Suzhou Future Membrane Technology Innovation Center, Suzhou, China. ✉e-mail: duanjingui@njtech.edu.cn; bjunfeng@njtech.edu.cn

restricting scalability and industrial adoption. Highlight the need for developing adsorbents with engineered pore geometries and optimized binding affinities—innovations critical to achieving highly pure $C_2H_2$.

Porous coordination polymers (PCPs), also known as metal-organic frameworks (MOFs), have emerged as a versatile platform for light hydrocarbon separations, owing to their precisely engineered pore geometries[9–11] and chemical functionalities[12–14] enabled by crystal engineering and isoreticular design principles[15–19]. The high quadrupole moment and π-electron density of C2 gases hydrocarbons drive strong interactions with polar binding sites or open metal centers in PCPs[20], leading to a typical adsorption sequence of $C_2H_2 > C_2H_4 > C_2H_6$[21]. However, to facilitate efficient and direct harvesting of highly pure $C_2H_4$ in the adsorption process, strategies involving supramolecular sites and structural dynamics have been developed for tuning the adsorption sequence[22–28], with notable success in $C_2H_4$ purification via partially reversed selectivity adsorption order ($C_2H_2 > C_2H_6 > C_2H_4$)[29–31]. However, due to its high quadrupole moment and small kinetic diameter, $C_2H_2$ remains the most strongly adsorbed species in most porous systems. By utilizing a perfluorinated narrow channel, a reversed adsorption order of C2 hydrocarbons has been observed in Zn-FBA; however, the negligible uptake differences between $C_2H_6$-$C_2H_2$ and $C_2H_4$-$C_2H_2$ render $C_2H_2$ purification infeasible from C2 ternary mixtures[32]. In other words, neither a function-driven approach nor a finely tuned structural design of PCPs has successfully achieved electronic-grade $C_2H_2$ purification from ternary C2 mixtures in one-step under industrially relevant conditions.

In this work, we address the challenge of purifying $C_2H_2$ from ternary C2 mixtures by adopting a counterintuitive strategy: weakening the host-guest interactions with $C_2H_2$ while enhancing the binding with its saturated counterparts, $C_2H_6$ and $C_2H_4$. To achieve this, we developed an adsorption-steering methodology utilizing a MOF-5 analog platform ($Zn_4O(PyC)_3$, PyC = 4-pyrazolecarboxylate) (Fig. 1a)[33,34]. Through incorporation of inert functional groups (trifluoromethyl and methyl groups) to the PyC ligand, we synthesized a series of MOF-5 analogs and realized several synergistic objectives: (1) maintaining a high surface area (>1000 $m^2\,g^{-1}$) to ensure adequate adsorption capacity; (2) strengthening van der Waals interactions with hydrogen-rich $C_2H_6$ and $C_2H_4$ while simultaneously suppressing the affinity for $C_2H_2$; and (3) enhancing the water-stability of the framework through steric protection of the metal nodes by hydrophobic methyl groups. Of them, **NTU-98**, featuring two methyl groups on the PyC ligand, represents a robust framework and exhibits a fully inversed adsorption order ($C_2H_6 > C_2H_4 > C_2H_2$), enabling the direct harvesting of high-purity $C_2H_2$ (>99.99%) from C2 ternary mixtures at 298 K. Density functional theory calculations and gas-loaded crystallographic analyses reveal that in **NTU-98**, the abundant methyl groups within the small cage, along with the methyl-decorated nitrogen/oxygen adsorption corners in the large cage, provide stronger interactions with the more saturated $C_2H_6$ and $C_2H_4$, while concurrently diminishing the affinity for $C_2H_2$. This behavior contrasts with other analogs with single methyl or trifluoromethyl group on the ligand, which exhibit significantly limited separation efficiency.

## Results
### Structure and adsorption properties
Colorless transparent crystals of **NTU-96**, **NTU-97** and **NTU-98** were obtained through solvothermal reactions of zinc nitrate hexahydrate and 4-PyC derivatives (TFPC, MePC and DiMePC, respectively) (Fig. 1b–d). Single crystal X-ray diffraction results demonstrated that **NTU-96** and **NTU-97** both crystallize in space group $Fm\text{-}3m$, while **NTU-98** crystallizes in $F\text{-}43m$ (Supplementary Table 1). The similar coordination properties of pyrazolate and carboxylate make these structures analogous to that of MOF-5, featuring a classical node of $Zn_4O$ (Supplementary Figs. 1–5). But, differently, the coordination of the two N

sites of 4-PyC derivatives greatly constraints the rotational freedom of the ligand, yielding the three-dimensional framework with two different cages in **NTU-96** to **NTU-98**. Upon ligand functionalization with trifluoromethyl, methyl, and two methyl groups, the internal diameter of the smaller cubic cage slightly contracts, shrinking from the original 8.0 Å (in $Zn_4O(PyC)_3$) to 5.8, 6.0, and 5.6 Å, respectively (Fig. 1b–d, Supplementary Fig. 6). Note that, although $Zn_4(\mu_4\text{-}O)(DiMePC)_3$ was previously investigated for the capture of nerve agents and mustard gas[35], the reported structure is not exactly the same as that of **NTU-98** (with different crystal symmetries), and the functional implications of the two methyl groups in modulating host–guest interactions remain unexplored. Phase purity of **NTU-96**, **NTU-97** and **NTU-98** were confirmed by the powder X-ray diffraction (PXRD) analysis (Supplementary Figs. 7–9). Thermogravimetric analysis (TGA) of as-synthesized and activated samples confirm that these materials are thermally stable up to 400 °C (Supplementary Figs. 10–15).

$N_2$ (77 K) and $CO_2$ (195 K) adsorption isotherms were collected to evaluate the permanent microporosity of **NTU-96** to **NTU-98**, respectively. Type-I adsorption isotherms were observed without any hysteresis (Supplementary Figs. 16, 17). **NTU-96** and **NTU-97** exhibit high $N_2$ uptake, up to 250.6 and 320.3 $cm^3\,g^{-1}$, respectively. However, the $N_2$ uptake of **NTU-98** (44.5 $cm^3\,g^{-1}$) is much lower than that of the other two, but it has comparable $CO_2$ uptake (240.3 $cm^3\,g^{-1}$). This is likely due to the blocking effect of aggregated $N_2$ at very low temperature. Based on the $CO_2$ isotherms, Brunauer-Emmet-Teller (BET) surface areas of the three were calculated to be ~1150, 1470 and 1245 $m^2\,g^{-1}$, respectively. The derived pore size distribution centers at 8.8 Å in **NTU-96**, 9.6 Å in **NTU-97** and 8.4 Å in **NTU-98**, which are all comparable to the geometrically determined values of the cavities based on the crystal structures.

To evaluate the role of nanopores in PCPs, single-component adsorption isotherms of $C_2H_2$, $C_2H_4$, and $C_2H_6$ were collected at 273–308 K, along with data for the precursor $Zn_4O(PyC)_3$ (Fig. 2, Supplementary Figs. 18–53, and Supplementary Table 2). $Zn_4O(PyC)_3$ exhibits a preference for $C_2H_6$ adsorption, with nearly overlapping adsorption isotherms for $C_2H_2$ and $C_2H_4$. This observation correlates with the zero coverage isosteric heat of adsorption ($Q_{st}$) values ($C_2H_6$: 25 $kJ\,mol^{-1}$; $C_2H_4$: 22.6 $kJ\,mol^{-1}$ and $C_2H_2$: 23.8 $kJ\,mol^{-1}$). In contrast, **NTU-96** demonstrates a completely reversed adsorption order for these gases. The $Q_{st}$ values for **NTU-96** are significantly higher, measuring 29.4 $kJ\,mol^{-1}$ for $C_2H_6$, 27.7 $kJ\,mol^{-1}$ for $C_2H_4$, and 26.5 $kJ\,mol^{-1}$ for $C_2H_2$, all exceeding those of $Zn_4O(PyC)_3$ (Table 1 and Supplementary Figs. 54–66). The elevated $Q_{st}$ for $C_2H_2$ suggests that the polar trifluoromethyl groups have a synergistic effect with some of the original adsorption sites rather than interfering with or shielding the adsorption sites. However, the uptake difference between $C_2H_4$ and $C_2H_2$ remains relatively modest, at 6.7 $cm^3\,g^{-1}$ at 298 K and 50 kPa. Upon substituting the trifluoromethyl with methyl group, **NTU-97** also exhibits a completely reversed adsorption order for C2 hydrocarbons. This alteration is accompanied by a moderate increase in the uptake difference between $C_2H_4$ and $C_2H_2$, recorded at 10.3 $cm^3\,g^{-1}$, despite a significantly larger uptake difference of 31.3 $cm^3\,g^{-1}$ between $C_2H_6$ and $C_2H_2$. Importantly, **NTU-98**, which contains two implanted methyl groups, demonstrates not only a fully inversed adsorption order, but also steeper uptake profiles for all three gases (Fig. 2a–c). When compared to benchmark materials, **NTU-98** exhibits the largest uptake difference for $C_2H_4$ and $C_2H_2$ (11.3 $cm^3\,g^{-1}$), while its uptake difference for $C_2H_6$ and $C_2H_2$ (26.5 $cm^3\,g^{-1}$) is only slightly smaller than that of **NTU-97** (31.3 $cm^3\,g^{-1}$) (Fig. 2d, e and Supplementary Table 3). Additionally, the $Q_{st}$ values for **NTU-98** are 32.2 $kJ\,mol^{-1}$ for $C_2H_6$ and 27.3 $kJ\,mol^{-1}$ for $C_2H_4$, both of which are marginally higher than those for **NTU-97** (28 and 25 $kJ\,mol^{-1}$, respectively). However, these values are still lower than that of Zn-FBA (42.8 and 39.8 $kJ\,mol^{-1}$), indicating a reduced energy requirement during cyclic operations. Considering the emerging adsorption order and systematically tuned uptake

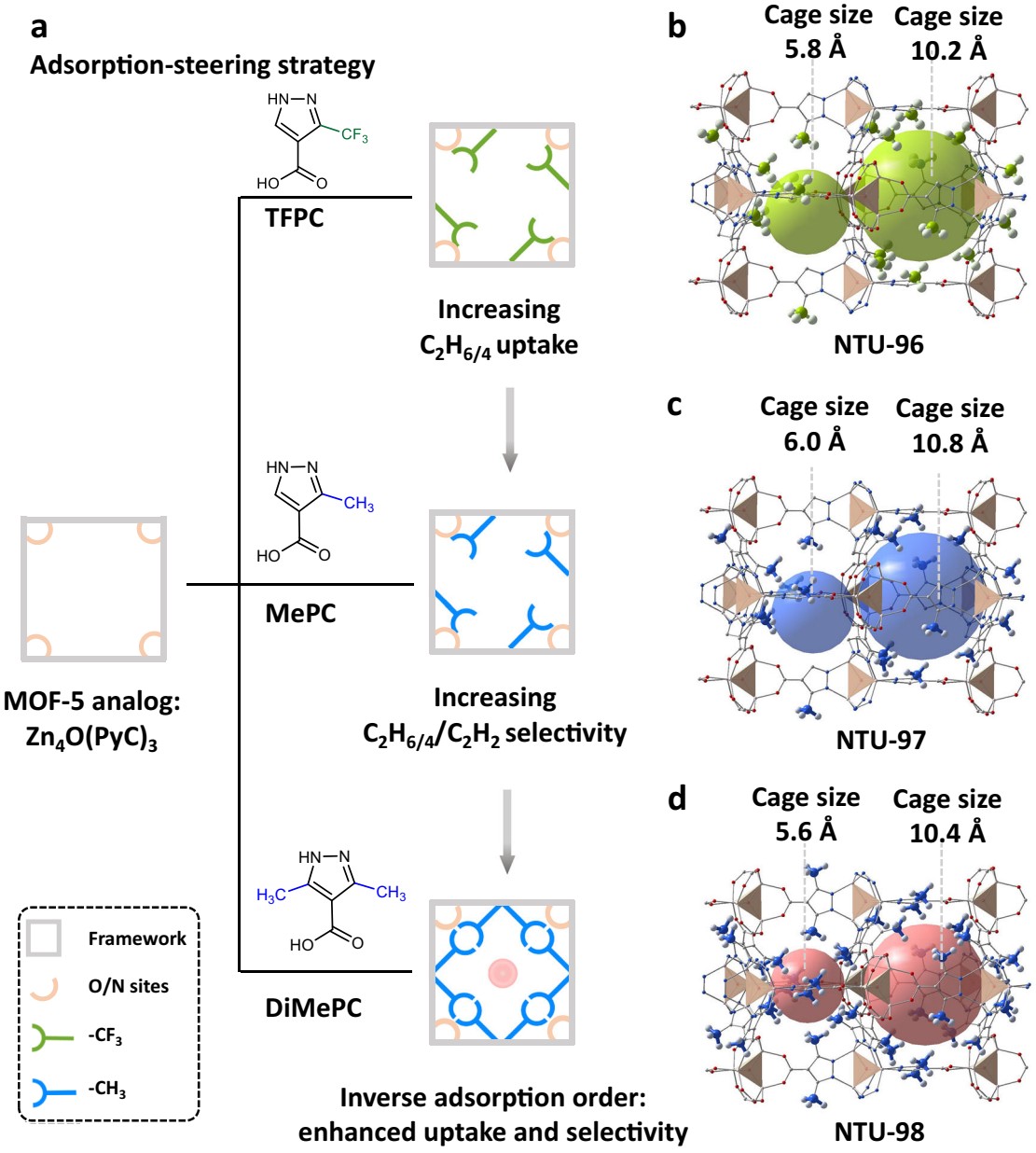

**Fig. 1 | Adsorption-steering strategy in the highly porous Zn₄O(PyC)₃ platform.**
**a** Systematic implantation of inert groups with varying types and loading amounts for enhanced $C_2H_{6/4}$ uptakes and $C_2H_{6/4}/C_2H_2$ selectivities. **b**–**d** Corresponding structures: **NTU-96, NTU-97** and **NTU-98**, respectively. Color code: Zn₄O, brown tetrahedra; C, gray; O, red; N, blue; H, white; F, cyan; void space, highlighted in green, blue and light red; gray row, fine-tuned structures.

differences, ideal adsorbed solution theory (IAST)[36–38] was applied to calculate the adsorption selectivity (Fig. 2f, Supplementary Figs. 67–71). **NTU-98** demonstrated exceptional adsorption selectivity for equimolar $C_2H_4/C_2H_2$ and $C_2H_6/C_2H_2$ mixtures, achieving values of 1.62 and 2.62, respectively, at 298 K and low pressure. These selectivities further increased at 273 K, reaching 1.95 ($C_2H_4/C_2H_2$) and 3.98 ($C_2H_6/C_2H_2$), underscoring its temperature-dependent performance. In contrast, **NTU-96** (1.25 and 1.88), **NTU-97** (1.51 and 2.32), and Zn₄O(PyC)₃ (0.93 and 1.09) exhibited markedly lower selectivities under identical conditions. The results highlight **NTU-98** as a superior adsorbent for the challenging separation of $C_2H_2$ from $C_2H_6$ and $C_2H_4$-containing mixtures.

## Adsorption mechanism studies

To elucidate the molecular interactions between **NTU-98** and C2 hydrocarbons, we conducted dispersion-corrected density functional

theory (DFT-D) calculations. Our analysis reveals that the primary adsorption sites for all three C2 hydrocarbons are localized at the center of the small cage and the corners of the large cage within the **NTU-98** framework. In Fig. 3, we show schematically the DFT-D-optimized adsorption structures of the three gas molecules at various adsorption sites. Within the small cage, the two $H_{C2H2}$ interact with two $C_{CH3}$ from DiMePc, forming C–H···C interactions with distances ranging from 3.253 to 3.353 Å (Fig. 3a). Additionally, a hydrogen bond is established between a $H_{CH3}$ and a $C_{C2H2}$ at a distance of 3.189 Å. Interestingly, four hydrogen bonds are observed between $H_{C2H4}$ and $C_{CH3}$ from DiMePc, with bond distances varying from 3.119 to 3.378 Å, coupled with a hydrogen bond between $H_{CH3}$ and $C_{C2H4}$ at 3.469 Å. In addition, due to the increased $H^{δ+}$ on $C_2H_6$, six hydrogen bonds are formed with the $C_{CH3}$ groups, with bond lengths ranging from 3.269 to 3.483 Å. The increased number of hydrogen bonds, although each has slightly different bond length, clearly indicates progressively stronger

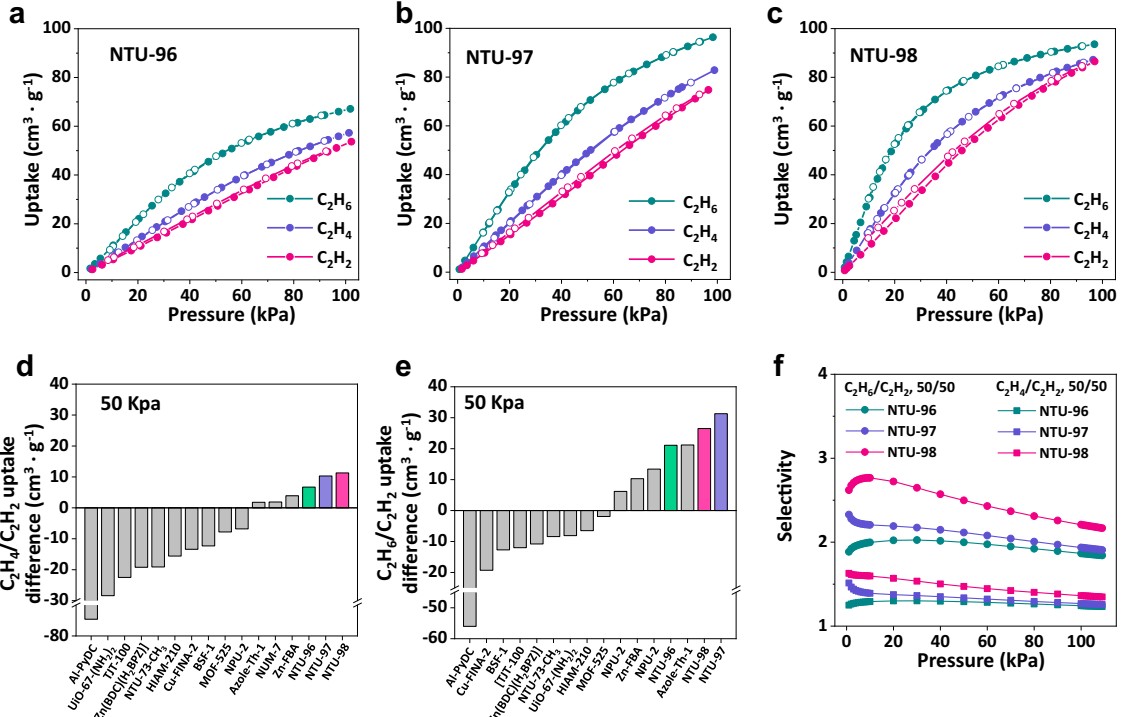

**Fig. 2 | Gas adsorption properties.** $C_2H_2$, $C_2H_4$ and $C_2H_6$ adsorption isotherms of **a** **NTU-96**, **b** **NTU-97** and **c** **NTU-98** at 298 K, respectively. Comparison of **d** $C_2H_4$/ $C_2H_2$ and **e** $C_2H_6$/$C_2H_2$ uptake difference of the benchmark PCPs at 298 K, 50 kPa. Values for literature materials are taken from refs: Al-PyDC[43], Uio-67-(NH$_2$)$_2$[31], TJT-100[29], [Zn(BDC)(H$_2$BPZ)].4H$_2$O[44], NTU-73-CH$_3$[45], HIAM-210[46], Cu-FINA-2[47], BSF-1[48], MOF-525[49], NPU-2[30], Azole-Th-1[25] and NUM-7[50]. **f** IAST selectivity of $C_2H_6$/$C_2H_2$ (1/ 1, v/v) and $C_2H_4$/$C_2H_2$ (1/1, v/v) at 298 K.

host–guest interactions for $C_2H_4$ and especially $C_2H_6$. Owing to the asymmetric nature of the ligand, the metal nodes within the framework can exhibit mixed coordination environments, featuring both Zn–N and Zn–O bonds. To accurately model the host-guest interactions, we defined two representative configurations for the large cage corners, differentiated by the identity of the coordinating atom (either nitrogen or oxygen). In both cases, $C_2H_6$ and $C_2H_4$ form more hydrogen bonds and C–H···π interactions with DiMePc than $C_2H_2$ (Fig. 3b, c). For comparison, we also performed calculations on the C2 hydrocarbon adsorption in the parent framework Zn$_4$O(PyC)$_3$. In the absence of functional methyl groups, adsorption primarily occurs at (i) the window aperture bridging small and large cages, and (ii) the N/O-defined adsorption corners. Meanwhile, the number of hydrogen bonds at N/O corners increases with the number of hydrogen atoms on the guest molecule ($C_2H_6$ > $C_2H_4$ > $C_2H_2$). However, at the window aperture, all three C2 species form nearly the same number of hydrogen bonds ($C_2H_2$: 8, $C_2H_4$: 7 and $C_2H_6$: 8), suggesting non-discriminative binding (Supplementary Fig. 72). This contrast— between site-specific selectivity and aperture-driven uniformity— explains the diminished C2 selectivity of the parent framework. These observations are further evident from the calculated static gas binding energies at various adsorption sites (Table 2), which are also fully consistent with the experimental $Q_{st}$ values.

To validate the computational modeling results, we further experimentally determined the gas-loaded crystal structures by single-crystal X-ray diffraction. Due to the high crystal symmetry, accurately resolving the orientations of the ligand and guest gas molecules is challenging, resulting in apparent orientational disorder in the experimental structures. Nevertheless, the experimental results confirm that the three primary adsorption sites—the center of small cage and the corners of large cage featuring N or O atoms—closely match the predictions from the modeling. Consistent with the calculations, gas molecules with higher $H^{\delta+}$ content exhibit stronger interactions

(Supplementary Fig. 73), and the experimentally observed hydrogen bond lengths are in close agreement with the calculated values. The adsorption kinetics are of the same order of magnitude, indicating a limited influence on the overall selectivity (Supplementary Fig. 73). Therefore, the high selectivity of **NTU-98** for $C_2H_4$ and $C_2H_6$ over $C_2H_2$ arises from the synergistic effect of strategically positioned methyl groups within both cages and the N/O-decorated adsorption sites. These features collectively enhance the cooperative interactions with the more saturated $C_2H_4$/$C_2H_6$, while simultaneously diminishing the affinity toward the more polarizable $C_2H_2$. This molecular-level design leverages intrinsic differences in C–H bond acidity across the C2 hydrocarbon series.

**Dynamic breakthrough experiments**

Considering the distinct adsorption order, breakthrough experiments were carried out on the three PCPs. After initial activation, the three samples were loaded into a column, followed by further activation. The system was flushed with He until no other signal was detected. Then the corresponding feed gas was introduced into the packed column. The clear time intervals indicate that **NTU-98** can effectively separate the binary mixtures of $C_2H_6$/$C_2H_2$ (v/v, 1/1) and $C_2H_4$/$C_2H_2$ (v/v, 1/1) (Fig. 4a, b and Supplementary Figs. 75, 76). After changing the feed gas to the C2 ternary mixtures, $C_2H_2$ eluted first out of the fixed bed of **NTU-98** at 56.5 min·g$^{-1}$, and the $C_2H_4$ breakthrough the column at 57.6 min·g$^{-1}$, followed by $C_2H_6$ at 67.1 min·g$^{-1}$, well matching the inversed adsorption behavior of **NTU-98**. With reference to the higher ratio of $C_2H_2$ during the deep purification step, the feed gas was then change to 90/9/1 ($C_2H_2$/$C_2H_4$/$C_2H_6$: v/v/v)[39,40]. Consistently, same elution sequences were observed, of which $C_2H_2$ with high purity (>99.99%) was detected at about 37.2 min·g$^{-1}$, followed by $C_2H_4$ at 41.6 min·g$^{-1}$, and $C_2H_6$ at 48.8 min·g$^{-1}$, the corresponding $C_2H_2$ yield is 8.64 mL g$^{-1}$ (Fig. 4c). Despite extensive research in PCP chemistry, the direct one-step purification of $C_2H_2$ with high purity from ternary C2

**Table 1 | Summarized surface areas, uptakes, adsorption heats and selectivities of the PCPs**

| PCPs | BET(m² g⁻¹) | Uptake (cm³ g⁻¹)50 kPa, 298 K | | | Qst (kJ mol⁻¹) | | | Selectivity (v/v = 1:1)298 K | |
|---|---|---|---|---|---|---|---|---|---|
| | | C2H2 | C2H4 | C2H6 | C2H2 | C2H4 | C2H6 | C2H4/C2H2 | C2H6/C2H2 |
| Zn₄O(PyC)₃ | 1350 | 39.9 | 40.6 | 61.5 | 23.8 | 22.6 | 25.1 | 0.93 | 1.09 |
| **NTU-96** | 1150 | 26.7 | 33.4 | 47.8 | 26.5 | 27.1 | 29.0 | 1.25 | 1.88 |
| **NTU-97** | 1470 | 38.5 | 48.8 | 69.8 | 23.3 | 25.0 | 28.0 | 1.51 | 2.32 |
| **NTU-98** | 1245 | 53.5 | 64.8 | 80.0 | 23.5 | 27.3 | 32.2 | 1.62 | 2.62 |

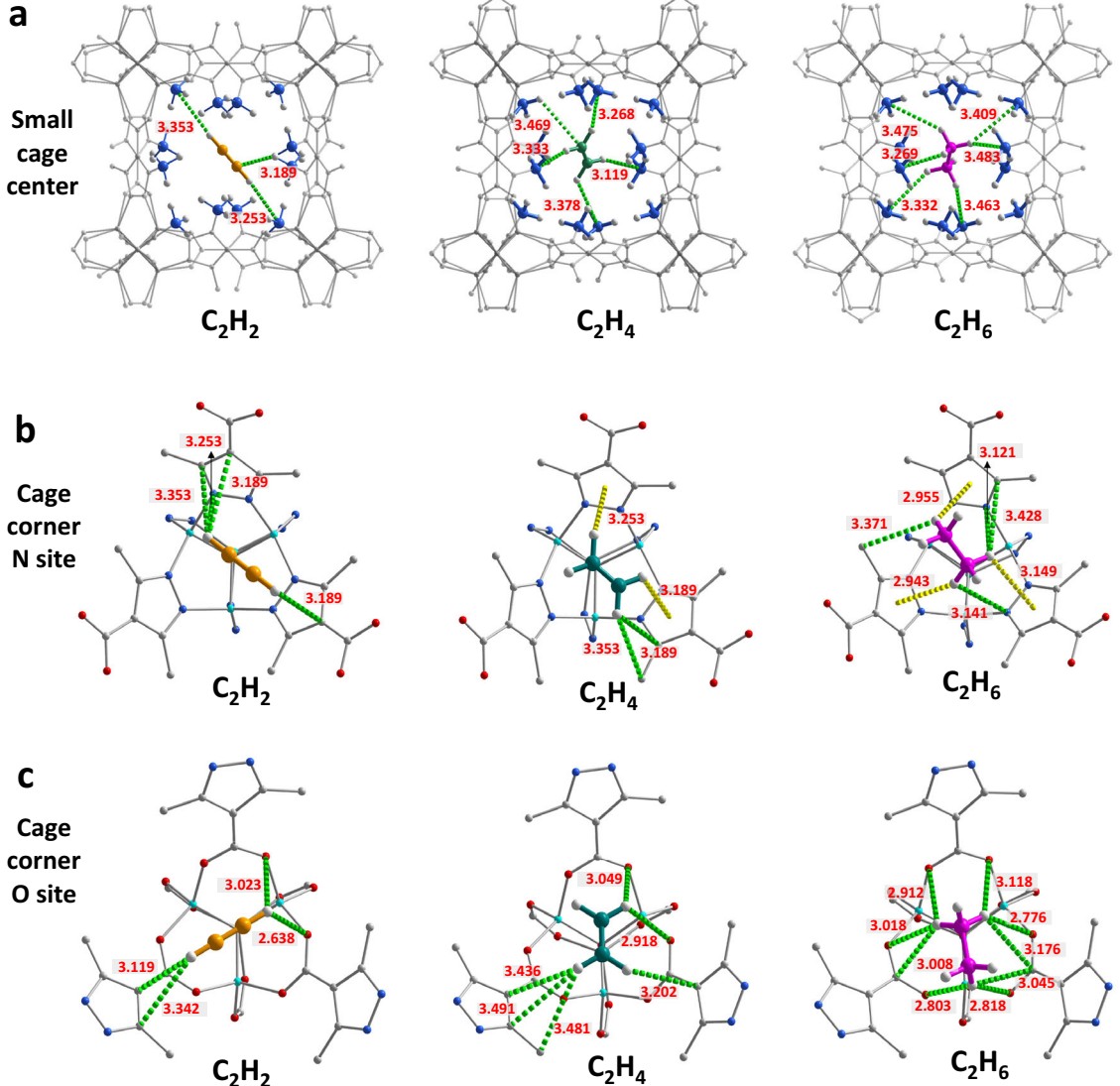

**Fig. 3 | DFT-D calculated host-guest interactions of NTU-98. a** C2 hydrocarbons located at the small cage center, **b** at the cage corner N site, and **c** at the cage corner O site. Color code: Zn, light green; C, gray; O, red; N, blue; H, white; C₂H₂: orange; C₂H₄: dark green; C₂H₆, purple; The green and yellow dashed lines indicate hydrogen bonding and C-H⋯π interaction, respectively.

mixtures remains, to the best of our knowledge, unreported. In comparison, **NTU-96** can also separate these ternary mixtures, but the separation time for harvesting of electronic grade C₂H₂ is short (Supplementary Figs. 77, 78).

For practical gas separation applications, scale-up synthesis and water stability of the adsorbent material are also two important factors to consider[41]. We found that rapid, scaled up synthesis of **NTU-98** can be readily achieved. By adding certain amount of NH₃·H₂O into the

stirring solution (400 mL scale) of the corresponding reactants, ~25 g of **NTU-98** can be obtained in 10 min, making the space-time yield as high as about 9000 kg/m³/day (Fig. 4d). Phase purity and porosity of the large-scale synthesized **NTU-98** were validated by PXRD and gas adsorption isotherms (Supplementary Fig. 79). Importantly, large-scale synthesized **NTU-98** demonstrates nearly the same separation performance for C2 hydrocarbons (Fig. 4e). Furthermore, we found that **NTU-98** exhibits excellent chemical stability. Even after

## Table 2 | DFT-D calculated static binding energies at various adsorptions sites (kJ mol$^{-1}$)

| | Cage corner O site | | | Cage corner N site | | | Cage window | | | Small cage center | | |
| --- | --- | --- | --- | --- | --- | --- | --- | --- | --- | --- | --- | --- |
| | $C_2H_2$ | $C_2H_4$ | $C_2H_6$ | $C_2H_2$ | $C_2H_4$ | $C_2H_6$ | $C_2H_2$ | $C_2H_4$ | $C_2H_6$ | $C_2H_2$ | $C_2H_4$ | $C_2H_6$ |
| $Zn_4O(PyC)_3$ | 22.4 | 24.3 | 24.5 | 21.0 | 22.7 | 23.6 | 25.4 | 22.7 | 24.2 | | | |
| **NTU-98** | 23.3 | 25.8 | 27.4 | 23.6 | 26.4 | 28.0 | | | | 24.5 | 27.5 | 30.7 |

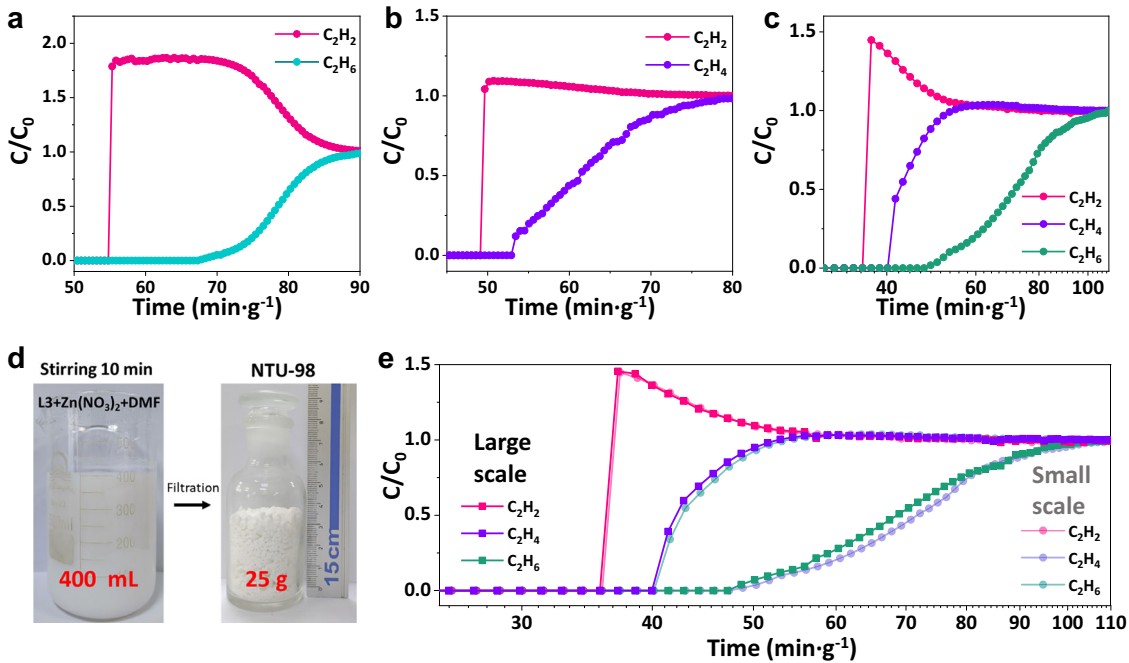

**Fig. 4 | Breakthrough experiments.** Breakthrough curves of **NTU-98** for **a** $C_2H_6$/$C_2H_2$ (1/1, v/v, 0.9 mL·min$^{-1}$), **b** $C_2H_4$/$C_2H_2$ (1/1, v/v, 0.9 mL·min$^{-1}$), **c** $C_2H_2$/$C_2H_4$/$C_2H_6$ (90/9/1,v/v/v, 2 mL·min$^{-1}$). **d** Photos of the large-scale synthesis for **NTU-98** (L3 = DiMePC ligand; DMF = Dimethylformamide). **e** Comparisons of breakthrough curves of small-scale and large-scale synthesized **NTU-98** for $C_2H_2$/$C_2H_4$/$C_2H_6$ (90/9/1, v/v/v, 2 mL·min$^{-1}$) mixtures. The temperature and pressure for all breakthrough experiments is 298 K and 1 bar.

immersion in solutions with pH ranging from 2 to 12 for one week, the sample largely retained its crystallinity and gas uptake capacity (Supplementary Fig. 80). Such high-water stability is fully expected because the two methyl groups on each ligand effectively shield the Zn-O coordination bonds from hydrolysis. Thanks to its stability, **NTU-98** does not suffer any notable performance loss during cycling breakthrough experiments (Supplementary Fig. 81).

## Discussion

Addressing the challenge of directly harvesting high-purity $C_2H_2$ from ternary C2 mixtures, we present an adsorption-steering strategy that optimizes ensemble host–guest interactions in a family of porous crystals. NTU-98, featuring dual methyl groups positioned at adsorption corners, exhibits a fully inverted adsorption hierarchy ($C_2H_6 > C_2H_4 > C_2H_2$). Gas-loaded crystallography and DFT-D analyses reveal that the synergistic interplay between sterically placed methyl groups and N/O-functionalized nodes enhances binding toward saturated hydrocarbons while diminishing affinity for polarizable $C_2H_2$. Of them, NTU-98 combines high structural robustness and synthetic scalability, enabling direct production of electronic-grade $C_2H_2$ (>99.99%) from ternary C2 feeds in one step under ambient conditions, which has not been demonstrated in earlier PCP studies. This design strategy is broadly applicable to other porous frameworks, providing a framework for exploring inverse adsorption hierarchies and advancing adsorbents for challenging separations.

## Methods

### Synthesis of NTU-96

5-(trifluoromethyl)-1H-pyrazole-4-carboxylic acid (TFPC) (4 mg, 0.022 mmol) and Zn(NO$_3$)2•6H$_2$O (12 mg, 0.04 mmol) were added to the 1 mL mixed solvent of DMA/H$_2$O (v/v, 1/1), sonicated and placed into a 10 mL glass bottle, and heated in an oven at 120 °C for 48 h. After the end of the reaction, colorless crystals are obtained. The NTU-96 crystals are washed with DMA and stored dry at room temperature.

### Synthesis of NTU-97

3-methyl-1H-pyrazole-4-carboxylic acid (MePC) (4 mg, 0.032 mmol) and Zn(NO$_3$)2•6H$_2$O (20 mg, 0.07 mmol) were added to the 1 mL mixed solvent of DEF/H$_2$O (v/v, 3/1), sonicated, and transferred to a 10 mL glass bottle. Subsequently, the glass bottles were heated in an oven at 85 °C for 48 h. After the end of the reaction, colorless crystals were obtained. The NTU-97 crystals are washed with DEF and stored dry at room temperature.

### Synthesis of NTU-98 (small-scale)

3,5-Dimethyl-1H-pyrazole-4-carboxylic acid (DiMePC, 14 mg, 0.1 mmol) and zinc nitrate hexahydrate (Zn(NO$_3$)$_2$·6H$_2$O, 27 mg, 0.09 mmol) were dissolved in 1 mL of N,N'-dimethylformamide (DMF) with the aid of sonication. The solution was then transferred into a 10 mL glass vial and heated at 120 °C for 24 h. Upon cooling to room temperature, colorless crystals of NTU-98 were obtained. The crystals were washed with fresh DMF and dried under ambient conditions.

### Synthesis of NTU-98 (large-scale)

For scale-up, the corresponding reactants (DiMePC: 14 g, 0.1 mol and $Zn(NO_3)_2 \cdot 6H_2O$: 27 g, 0.09 mol) were dissolved in 400 mL of DMF with stirring, followed by the addition of an appropriate amount of aqueous ammonia ($NH_3 \cdot H_2O$). Crystallization occurred rapidly, yielding ~25 g of NTU-98 within 10 min.

### Gas adsorption measurements

Single gas adsorption isotherms were performed on a Belsorp volumetric adsorption instrument (BEL Japan Corp.). In the sorption measurements, ultra-high-purity grade gases of $C_2H_2$, $C_2H_4$, $C_2H_6$ and $CO_2$ were used throughout the adsorption experiments.

### Dynamic breakthrough experiments

Breakthrough experiments were performed on the Beifang Gaorui CT-4 system. The initial activated samples were tightly packed into a stainless-steel column ($\varphi = 0.40$ cm, L = 30 cm). Then, the column was activated under vacuum at the corresponding temperature and then swept with helium (He) flow to remove impurities. Until no signal was detected, the gas flow was dosed into the column. Breakpoints were determined by gas chromatography. Between cycling experiments, regeneration can be achieved under high vacuum at 393 K for half-hour. Pressure of the feed gas is 1 bar. For breakthrough experiments, the mixtures of $C_2H_6/C_2H_2$, $C_2H_4/C_2H_2$ and $C_2H_6/C_2H_4/C_2H_2$ were obtained by utilizing a premix gas cylinder.

## Data availability

The crystal data generated in this study have been deposited in the Cambridge Crystallographic Data Centre (CCDC) under accession code 2434597-2434601 and 2434605 [https://www.ccdc.cam.ac.uk/structures]. Source data of the sorption tests, gas adsorption enthalpies, selectivities and break through tests that support the findings of this study are provided as a SourceData file (ref. 42). Source data are provided with this paper.

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

## Acknowledgements
This work was supported by the National Natural Science Foundation of China (22171135, 22471235, both to J.D.; 22271150 to J.B.), the Out-standing Young Scientist of Natural Science Foundation of Xinjiang Uygur Autonomous Region of China (2025D01E06, to J.D.), the Natural Science Foundation of Jiangsu Province (BK20231269, to J.D.), the State Key Laboratory of Materials-Oriented Chemical Engineering (SKL-MCE-23A18, to J.D.), and the Jiangsu Future Membrane Technology Innovation Center (BM2021804, to J.D.). We gratefully acknowledge Dr. Wei Zhou for providing valuable assistance with the DFT calculations and helpful suggestions during manuscript revision.

## Author contributions
J.D. and J.B. conceived this project. M.Z. and Y.F. performed the synth-esis experiments. J.D. and M.Z. analyzed the experimental data. J.B. and J.D. drafted the paper with support from M.Z. All authors discussed the results and contributed to the preparation of the final manuscript.

## Competing interests
The authors declare no competing interests.
