## [Transparent Peer Review file · Nature Communications]

Fully inverse adsorption enables one-step high-purity C₂H₂ separation from ternary C₂ mixtures in a robust porous crystal

Corresponding Author: Professor Jingui Duan

Version 1:

Reviewer comments:

Reviewer #1

(Remarks to the Author)

In this work, the authors developed an adsorption-steering strategy for constructing a robust MOF with a fully inverted C₂ adsorption hierarchy, enabling for the first time the direct production of electronic-grade C₂H₂ (>99.99%) from ternary C₂ feeds in one step at ambient conditions. This is an excellent and innovative work that addresses a long-standing and industrially important challenge in the field of light hydrocarbon separations. The work is characterized by synthetic scalability, stability of the material, and a clear demonstration of its superior separation performance through breakthrough experiments. More importantly, the underlined mechanism was illustrated by thorough DFT calculations and crystallographic evidence, both are impactful. The manuscript is well-written, the data are comprehensive and convincing, and the conclusions are supported. Here, I only offer some minor suggestions:

1. The authors might consider commenting on the effect of competitive adsorption by CO₂ on the performance of NTU-98. This would strengthen the practical relevance.
2. The authors attribute the separation performance primarily to thermodynamic (host-guest interaction) factors, supported by Qst and DFT calculations. However, given the reduced pore size in NTU-98, some contribution from kinetic sieving might be expected, especially between C₂H₂ and C₂H₄. A discussion clarifying whether diffusion rates contribute to the inverted selectivity would enrich the mechanistic understanding.
3. The breakthrough experiments clearly demonstrate the separation order. However, it would strengthen the work if the authors could extract and report productivity metrics from the breakthrough data — e.g., working capacities (mmol/g), dynamic selectivities, and specific productivity (mol C₂H₂/kg adsorbent) — to facilitate comparison with other benchmark materials.
4. Some of the figures, particularly Figure 3 with the DFT interactions and the breakthrough curves, would benefit from clearer, more detailed captions explaining the key features and highlighting the critical observations. Adding schematic labels for the cages and interaction sites in the figures would also help readers follow the discussion more easily.
5. While the writing is generally clear, there are a few minor grammatical errors and awkward phrases (e.g., “synthesized” instead of “synthesized”; “simplicity” instead of “simplicity”).

Reviewer #2

(Remarks to the Author)

In this manuscript, the authors reported three porous MOFs of NTU-96, NTU-97 and NTU-98 using the pyrazole carboxylate ligand functionalized with trifluoromethyl/methyl groups. NTU-98 shows unusual adsorption trend of C₂ hydrocarbons of C₂H₆ > C₂H₄ > C₂H₂, enabling direct production of ultrahigh-purity C₂H₂ (>99.99%) from ternary feeds at room temperature. Density functional theory calculations and gas-loaded crystallographic analyses reveal that the presence of methyl groups in NTU-98 enhances the host-guest interactions with the more saturated C₂H₄ and C₂H₆, while concurrently weaken the affinity for the more polarizable C₂H₂. Though the results are interesting, and both the structures and adsorption performance were well investigated, the meaning and purpose of this investigation is not clear, thus I cannot recommend it for publication based on the following reasons:

1. In petrochemical industry, it usually needs to obtain high-purity ethylene from a C₂H₆/ C₂H₄ / C₂H₂ three-component mixture. I wonder the application scenario .for getting high purity C₂H₂ from a C₂H₆/ C₂H₄ / C₂H₂ three-component

mixture?

2. In Figure 4a and 4b, the breakthrough curve of NTU-98 for a 1:1 mixture of C₂H₆/C₂H₂ or C₂H₄/C₂H₂ shows that C₂H₂ is adsorbed preferentially. However, in Figure 4c, C₂H₆ and C₂H₄ are adsorbed preferentially for a C₂H₆/ C₂H₄ / C₂H₂ three-component mixture, please give a reasonable explanation.

3. How about the breakthrough curves of NTU-96 and NTU-97 for a C₂H₆/ C₂H₄ / C₂H₂ three-component mixture?

Reviewer #3

(Remarks to the Author)

In this manuscript, the authors demonstrate a steric pore-environment engineering strategy that can realize one-step high-purity C₂H₂ separation from ternary C₂ mixtures in microporous crystals. In particular, NTU-98 fully reverses the adsorption trend of C₂ hydrocarbons (C₂H₆ > C₂H₄ > C₂H₂) and exhibits high IAST selectivities of C₂H₄/C₂H₂ and C₂H₆/C₂H₂ mixtures. The intrinsic mechanism for this separation performance was verified by DFT-D calculations and crystallographic analyses. This is an interesting piece of work, and it could be considered for publication in Nature Communications, if specific comments below can be well addressed.

1. On Page 4, Line 71-73, the authors claimed that "However, with the highest quadrupole moment, the smallest kinetic diameter, and the lowest polarizability, C₂H₂ remains the most preferentially adsorbed species across the most systems." However, it is generally accepted that high polarizability leads to enhanced host-guest interactions. Please explain why or correct this expression.

2. In the pore distribution in Figure S16, NTU-98 exhibited a significantly smaller micropore volume compared to NTU-96 and NTU-97. Whereas, the CO₂ uptake at low-pressure region is the second largest between the three analogues. Please check.

3. NTU-97 with single methyl groups shows better C₂H₆/C₂H₂ and C₂H₄/C₂H₂ separation performance than NTU-96 featuring single trifluoromethyl groups. The reason should be explained.

4. In Page 13, Line 212-214, the author states that "Due to the high crystal symmetry, in the experimental structures, there exist substantial orientational disorder of the ligand and the guest gas molecules." The logical connection between the high symmetry and observed disorder should be expressed as a contrast rather than causation.

5. What is the exact number of guest molecules within the small cage and large cage determined from the gas-loaded crystal structures obtained from the single-crystal X-ray diffraction? Is this result consistent with the uptake measured from the gas isotherms?

6. As shown in Figure S77, The PXRD patterns of the large-scale synthesized NTU-98 exhibits multiple extra peaks compared to that of the small-scale sample and the simulated XRD pattern. Does this phenomenon indicate the large-scale synthesized NTU-98 contain some impurities?

7. In the introduction, the authors mentioned this pore-environment engineering strategy enhances the framework's hydrophobicity and stability. No information on hydrophobicity was shown in the current version of manuscript. H₂O isotherms at various relative humidity or water contact angles should be added to illustrate the hydrophobicity of NTU-98.

8. In Figure 4a-4b, the breakthrough curves of the binary gases indicate C₂H₆ and C₂H₄ elutes out earlier than C₂H₂, which is contradictory with the result from isotherms and ternary breakthrough experiment. Please check.

9. In Page 9, Line 156, the sentence "However, these values yet still lower than that of Zn-FBA (42.8 and 39.8 kJ mol⁻¹)" lacks a missing predicate verb (e.g., 'are').

10. In Page 11, Line 189, "simpliticity" should be "simplicity".

11. Some formatting issues exist in the figures. For example, the y-axis title in Figure 2d-2e is written as "C₂H₄/C₂H₂ Uptake difference", while the labels in Figures 3b and 3c use "Cage Corner N site" and "Cage Corner O site". These should follow consistent capitalization rules (either title case or sentence case) throughout.

Version 2:

Reviewer comments:

Reviewer #1

(Remarks to the Author)

I am satisfied with the current version, which can be accepted.

Reviewer #2

(Remarks to the Author)

I recommend it for publication as it is now.

Reviewer #3

(Remarks to the Author)

The authors have fully responded to my comments and have revised the manuscript with high quality. The manuscript can be accepted in its current form.

Scientific comment from reviewers

Reviewer #1 (Remarks to the Author):

In this work, the authors developed an adsorption-steering strategy for constructing a robust MOF with a fully inverted C2 adsorption hierarchy, enabling for the first time the direct production of electronic-grade C₂H₂ (>99.99%) from ternary C2 feeds in one step at ambient conditions. This is an excellent and innovative work that addresses a long-standing and industrially important challenge in the field of light hydrocarbon separations. The work is characterized by synthetic scalability, stability of the material, and a clear demonstration of its superior separation performance through breakthrough experiments. More importantly, the underlined mechanism was illustrated by thorough DFT calculations and crystallographic evidence, both are impactful. The manuscript is well-written, the data are comprehensive and convincing, and the conclusions are supported. Here, I only offer some minor suggestions:

Answer:

We all appreciate your nice and positive comments.

1. The authors might consider commenting on the effect of competitive adsorption by CO₂ on the performance of NTU-98. This would strengthen the practical relevance.

Answer:

Thanks for your valuable suggestion.

CO₂ adsorption isotherm for NTU-98 has been newly collected at 298 K (see figure(a) below). The CO₂ uptake is consistently lower than that of C₂H₂ across the whole pressure range. This observation indicates that the impact of competitive CO₂ adsorption on the performance of NTU-98 for C₂H₂ purification from ternary C2 feeds is minimal.

Although CO₂ removal is not the primary objective of the current work, the process leverages the distinct adsorption behaviors governed by the ideal adsorbed solution theory (IAST). To achieve high-purity C₂H₂ separation from a C₂H₆/C₂H₄/C₂H₂/CO₂ mixture, we envision that a dual-bed system utilizing NTU-98 (A and B) can be employed. Initially, CO₂ elutes first from Bed A, followed by C₂H₂. Due to the pronounced difference in adsorption affinity between CO₂ and C₂H₂ on NTU-98, the desorption phase of Bed B yields a C₂H₂ product would contain only trace amounts of CO₂, ensuring high selectivity and purity.

2. The authors attribute the separation performance primarily to thermodynamic (host-guest interaction) factors, supported by Qst and DFT calculations. However, given the reduced pore size in NTU-98, some contribution from kinetic sieving might be expected, especially between C₂H₂ and C₂H₄. A discussion clarifying whether diffusion rates contribute to the inverted selectivity would enrich the mechanistic understanding.

Answer:

Thanks for your suggestion.

The adsorption kinetics of NTU-98 toward C₂ species were performed in collaboration with another laboratory, with the derived D_s/R^2 values following the trend C₂H₂ ($4.37 \times 10^{-3} \text{ s}^{-1}$) > C₂H₄ ($2.59 \times 10^{-3} \text{ s}^{-1}$) > C₂H₆ ($1.84 \times 10^{-3} \text{ s}^{-1}$). These values are of the same order of magnitude, indicating that diffusion kinetics have a limited influence on the overall selectivity. The data have been included in the Supplementary Information as Supplementary Fig. 74. The following sentence was added to the draft:

“The adsorption kinetics are of the same order of magnitude, indicating a limited influence on the overall selectivity (Supplementary Fig.74).”

Supplementary Fig.74. Diffusion rate of (a) C₂H₂ (b) C₂H₄ (c) C₂H₆ on NTU-98, fitted automatically with BEL-Dyna software according to the Crank theory. C, concentration; C⁰, initial concentration; C^e, concentration at equilibrium.

3. The breakthrough experiments clearly demonstrate the separation order. However, it would strengthen the work if the authors could extract and report productivity metrics from the breakthrough data — e.g., working capacities (mmol/g), dynamic selectivities, and specific

productivity (mol C₂H₂/kg adsorbent) — to facilitate comparison with other benchmark materials.

Answer:

Thanks for your nice suggestion.

1. Working capacity: this is typically defined for single-component gases; since our study focuses on binary and ternary gas adsorption separation, the working capacities were not calculated.
2. Dynamic selectivities: based on the breakthrough results with binary mixtures, dynamic selectivities were calculated. This result has been updated in SI.

Supplementary Fig.78. Calculated dynamic selectivities of C₂H₆/C₂H₂ and C₂H₄/C₂H₂ based on the breakthrough results with binary mixtures.

3. Specific productivity: for 90/9/1 (C₂H₂/C₂H₄/C₂H₆, v/v/v), C₂H₂ with a high purity (>99.99%) was detected at approximately 37.2 min g⁻¹, followed by C₂H₄ at 41.6 min g⁻¹ and C₂H₆ at 48.8 min g⁻¹. The calculated C₂H₂ yield is 8.64 mL g⁻¹. Accordingly, the following sentence has been added to the revised draft: “The corresponding C₂H₂ yield is 8.64 mL g⁻¹ .”

4. Some of the figures, particularly Figure 3 with the DFT interactions and the breakthrough curves, would benefit from clearer, more detailed captions explaining the key features and highlighting the critical observations. Adding schematic labels for the cages and interaction sites in the figures would also help readers follow the discussion more easily.

Answer:

Thanks for your suggestion.

We have revised Figure 3 to enhance clarity and readability. Schematic labels for the cages and interaction sites have been incorporated to help readers follow the discussion more easily.

Figure 3. DFT-D calculated host-guest interactions of NTU-98: **a** C_2 hydrocarbons inside the small cage center, **b** at the cage corner N site, and **c** at the cage corner O site. The green and yellow dash lines indicate the hydrogen bonding and C-H... π interaction, respectively.

5. While the writing is generally clear, there are a few minor grammatical errors and awkward phrases (e.g., “sythesized” instead of “synthesized”; “simplicty” instead of “simplicity”).

Answer:

We are very sorry for these typos. We have carefully corrected these errors and thoroughly rechecked the entire manuscript to ensure clarity and accuracy.

Reviewer #2 (Remarks to the Author):

In this manuscript, the authors reported three porous MOFs of NTU-96, NTU-97 and NTU-98 using the pyrazole carboxylate ligand functionalized with trifluoromethyl/methyl groups. NTU-98 shows unusual adsorption trend of C_2 hydrocarbons of $C_2H_6 > C_2H_4 > C_2H_2$, enabling direct production of ultrahigh-purity C_2H_2 (>99.99%) from ternary feeds at room temperature. Density functional theory calculations and gas-loaded crystallographic analyses reveal that the presence of methyl groups in NTU-98 enhances the host-guest interactions with the more saturated C_2H_4 and C_2H_6 , while concurrently weaken the affinity for the more polarizable C_2H_2 . Though the results are interesting, and both the structures and adsorption performance were well investigated, the meaning and purpose of this investigation is not clear, thus I cannot recommend it for publication based on the following reasons:

Answer:

Thank you for your positive feedback regarding our work, particularly your comments on the

interesting results and the thorough investigation of both the structures and adsorption performance.

Regarding the comment “the meaning and purpose of this investigation is not clear,” we have revised the manuscript to clarify this point. The revised text now reads: “The fully reversed adsorption trend of C2 hydrocarbons ($C_2H_6 > C_2H_4 > C_2H_2$) enables the direct production of highly pure C_2H_2 (over 99.99% purity) from ternary C2 feeds in a single step. This unprecedented finding is particularly remarkable given the extensive exploration of MOF chemistry to date.”

1. In petrochemical industry, it usually needs to obtain high-purity ethylene from a $C_2H_6/ C_2H_4 / C_2H_2$ three-component mixture. I wonder the application scenario for getting high purity C_2H_2 from a $C_2H_6/ C_2H_4 / C_2H_2$ three-component mixture?

Answer:

Thank you for your question. Please allow us to answer this question in two points.

1. The challenge and importance of high-purity C_2H_2 separation: While the purification of C_2H_2 from $C_2H_6/C_2H_4/C_2H_2$ mixtures has been extensively studied, the direct production of high-purity C_2H_2 from the same ternary mixture remains an unmet challenge. Unlike conventional adsorbents that preferentially capture C_2H_2 , achieving high-purity C_2H_2 requires a fully reversed adsorption hierarchy ($C_2H_6 > C_2H_4 > C_2H_2$)—a demanding criterion that has yet to be realized in a single-step separation process. In other words, obtaining high-purity C_2H_4 from a $C_2H_6/C_2H_4/C_2H_2$ three-component mixture is indeed important; however, this does not diminish the significance of producing high-purity C_2H_2 from the same mixture.
2. Highly pure C_2H_2 (>99.99%) is necessary in several advanced industries: a) Electronic-grade C_2H_2 is used in plasma-enhanced chemical vapor deposition to synthesize carbon films, graphene, and other nanostructured materials. b) As a key precursor, C_2H_2 enables the production of fine chemicals, pharmaceuticals (e.g., vitamin A), and high-value organic intermediates.

Given the challenge and applications, the development of adsorbents capable of selectively adsorbing C_2H_6 and C_2H_4 from C2 ternary mixture would represent a significant breakthrough in gas separation technology.

2. In Figure 4a and 4b, the breakthrough curve of NTU-98 for a 1:1 mixture of C_2H_6/C_2H_2 or C_2H_4/C_2H_2 shows that C_2H_2 is adsorbed preferentially. However, in Figure 4c, C_2H_6 and C_2H_4 are adsorbed preferentially for a $C_2H_6/ C_2H_4 / C_2H_2$ three-component mixture, please give a reasonable explanation.

Answer:

Thanks for pointing this out. We sincerely apologize for this serious mistake, which was a clear case of mislabeling. This important issue has now been corrected.

Updated Figure 4a-4b

3. How about the breakthrough curves of NTU-96 and NTU-97 for a $C_2H_6/C_2H_4/C_2H_2$ three-component mixture?

Answer:

Thank you for your suggestion.

For NTU-96, we newly collected breakthrough data for C_2H_6/C_2H_2 (1/1, v/v), C_2H_4/C_2H_2 (1/1, v/v), and the ternary mixture $C_2H_2/C_2H_4/C_2H_6$ (90/9/1, v/v/v). As shown in Supplementary Fig. 77, NTU-96 affords a shorter separation time for C_2H_2 ($2.1 \text{ min}\cdot\text{g}^{-1}$) than NTU-98 ($4.4 \text{ min}\cdot\text{g}^{-1}$).

The synthesis of NTU-97 is extremely sensitive, as even small deviations in the reaction parameters result in failure to obtain the desired product. After considerable effort, only about 50 mg of NTU-97 could be prepared, which is enough for adsorption measurements but far below the $\sim 2 \text{ g}$ typically required for breakthrough experiments. Moreover, NTU-97 is not considered the most promising material within this series.

Supplementary Fig.77. Breakthrough curves of NTU-96 (2.8107 g) at 298 K, 1 bar: (a) C_2H_6/C_2H_2 mixture (1/1, v/v, $0.9 \text{ mL}\cdot\text{min}^{-1}$), (b) C_2H_4/C_2H_2 mixture (1/1, v/v, $0.9 \text{ mL}\cdot\text{min}^{-1}$), (c) ternary mixtures $C_2H_2/C_2H_4/C_2H_6$ (90/9/1, v/v/v, $2 \text{ mL}\cdot\text{min}^{-1}$), (d) cycle breakthrough tests.

Reviewer #3 (Remarks to the Author):

In this manuscript, the authors demonstrate a steric pore-environment engineering strategy that can realize one-step high-purity C₂H₂ separation from ternary C₂ mixtures in microporous crystals. In particular, NTU-98 fully reverses the adsorption trend of C₂ hydrocarbons (C₂H₆ > C₂H₄ > C₂H₂) and exhibits high IAST selectivities of C₂H₄/C₂H₂ and C₂H₆/C₂H₂ mixtures. The intrinsic mechanism for this separation performance was verified by DFT-D calculations and crystallographic analyses. This is an interesting piece of work, and it could be considered for publication in Nature Communications, if specific comments below can be well addressed.

Answer:

We all appreciate your nice and positive comments.

1. On Page 4, Line 71-73, the authors claimed that “However, with the highest quadrupole moment, the smallest kinetic diameter, and the lowest polarizability, C₂H₂ remains the most preferentially adsorbed species across the most systems.” However, it is generally accepted that high polarizability leads to enhanced host-guest interactions. Please explain why or correct this expression.

Answer:

Thanks for your insight.

We agree with your opinion that high polarizability is typically advantageous for enhancing host-guest interactions through dispersion forces. However, in many porous frameworks, the preferential adsorption of C₂H₂ is primarily dominated by its significant quadrupole moment and small kinetic diameter, rather than its polarizability.

To avoid potential misunderstanding, we have revised the sentence on Page 4, Lines 71–73: “However, due to its high quadrupole moment and small kinetic diameter, C₂H₂ remains the most strongly adsorbed species in most porous systems.”

2. In the pore distribution in Figure S16, NTU-98 exhibited a significantly smaller micropore volume compared to NTU-96 and NTU-97. Whereas, the CO₂ uptake at low-pressure region is the second largest between the three analogues. Please check.

Answer:

Thanks for your nice question.

From a general theoretical perspective, the adsorption of CO₂ at low pressure is primarily influenced by the pore size (i.e., the binding strength), whereas at high pressure it is predominantly governed by the pore volume. CO₂ adsorption of NTU-98 follows this rule well.

Here are the detailed reasons:

1. Enhanced Low-Pressure CO₂ Uptake in NTU-98: The methyl groups may introduce narrowed pore environments that enhance CO₂ adsorption at low pressures.
2. Reduced High-Pressure CO₂ Uptake in NTU-98: The two methyl groups partially occupy the pore, reducing the available space for CO₂ adsorption at higher pressures where multilayer adsorption or pore filling becomes dominant.

Log-view of X-axis of Supplementary Fig.16.

3. NTU-97 with single methyl groups shows better C_2H_6/C_2H_2 and C_2H_4/C_2H_2 separation performance than NTU-96 featuring single trifluoromethyl groups. The reason should be explained.

Answer:

Thank you for your insightful question.

The trifluoromethyl ($-CF_3$) group is strongly electron-withdrawing due to the high electronegativity of fluorine atoms. This induces a partial positive charge on adjacent sites, enhancing specific interactions (e.g., hydrogen bonding or dipole interactions) with the relatively electron-rich C_2H_2 molecule. In contrast, the methyl ($-CH_3$) group is electron-donating and more hydrophobic, resulting in weaker or less specific interactions with C_2H_2 .

Incorporation of $-CH_3$ and $-CF_3$ groups into porous frameworks significantly influences gas adsorption behavior, though their effects vary depending on the gas species and functional group. Both modifications increase the adsorption enthalpies of C_2H_4 and C_2H_6 , indicating stronger host-guest interactions. However, while $-CF_3$ groups substantially enhance binding affinity for C_2H_2 , $-CH_3$ groups have minimal impact on C_2H_2 adsorption, suggesting stronger interactions between $-CF_3$ and C_2H_2 .

When comparing uptake capacities, the two functionalized frameworks show contrasting trends. NTU-96 (with $-CF_3$ groups) exhibits a uniform decrease in gas adsorption for all three hydrocarbons, with reductions of 13, 16, and 11 $cm^3 g^{-1}$ for C_2H_2 , C_2H_4 , and C_2H_6 , respectively. This indicates that although $-CF_3$ groups strengthen interactions, they also reduce pore accessibility, likely due to steric hindrance or pore blocking. Conversely, NTU-97 (with $-CH_3$ groups) shows selective enhancement: C_2H_2 uptake remains unchanged, while C_2H_4 and C_2H_6 adsorption increase by approximately 8 $cm^3 g^{-1}$ each. This selectivity aligns with adsorption enthalpy trends, confirming that $-CH_3$ groups preferentially enhance C_2H_4 and C_2H_6 uptake without affecting C_2H_2 .

Consequently, these differing adsorption profiles explain why NTU-97 outperforms NTU-96 in hydrocarbon separation. The $-CH_3$ groups in NTU-97 enhance C_2H_4 and C_2H_6 adsorption while leaving C_2H_2 uptake unaffected, improving selectivity for C_2H_4/C_2H_2 and C_2H_6/C_2H_2 separation. In contrast, $-CF_3$ groups in NTU-96 strengthen interactions with all three gases but reduce overall uptake, diminishing separation efficiency. Thus, methyl-functionalized NTU-97 emerges as the superior material for selective hydrocarbon capture, illustrating how subtle modifications in pore chemistry can fine-tune gas adsorption performance.

4. In Page 13, Line 212-214, the author states that “Due to the high crystal symmetry, in the experimental structures, there exist substantial orientational disorder of the ligand and the guest gas molecules.” The logical connection between the high symmetry and observed disorder should

be expressed as a contrast rather than causation.

Answer:

We appreciate the reviewer's insightful comment. We agree that high crystal symmetry does not directly cause orientational disorder but may complicate the precise determination of ligand and guest orientations in the experimental structure. We have revised the sentence accordingly to better reflect this relationship:

"Due to the high crystal symmetry, accurately resolving the orientations of the ligand and guest gas molecules is challenging, resulting in apparent orientational disorder in the experimental structures."

For the connection of each cluster, the direction of the ligand A and B is defined (see below left). But after rotating 90°, the view (determined by X-ray diffraction) of the ligand A and B is disordered. Similar behavior may also occur on the gases, as the symmetry of the crystal was strongly determined by cluster (contributing strongly diffraction). This is phenomenon and explanation are common in crystal chemistry (see *Inorg. Chem.* 2015. 54. 4279).

5. What is the exact number of guest molecules within the small cage and large cage determined from the gas-loaded crystal structures obtained from the single-crystal X-ray diffraction? Is this result consistent with the uptake measured from the gas isotherms?

Answer:

We thank the reviewer for this insightful question.

The uptake capacities estimated from the single-crystal structural analysis are in good agreement with those derived from adsorption isotherms. During the refinement of the gas-loaded single-crystal data, we found that a guest molecule occupancy ratio of approximately 1:4 (small pores to large pores) provides physically reasonable positioning of the guest molecules within the framework, as represented by the thermal ellipsoids. The specific occupancies obtained from crystallographic analysis and their comparison with isotherm-derived uptakes are summarized in the table below:

	C ₂ H ₂	C ₂ H ₄	C ₂ H ₆
Uptakes from isotherms	88 (cm ³ g ⁻¹)	87 (cm ³ g ⁻¹)	94 (cm ³ g ⁻¹)
Uptakes from crystals	77.7 (cm ³ g ⁻¹)	81.5 (cm ³ g ⁻¹)	90.1 (cm ³ g ⁻¹)
Ratios (small/large cage)	1/4	1/4.25	1/4.48

6. As shown in Figure S77, The PXRD patterns of the large-scale synthesized NTU-98 exhibits multiple extra peaks compared to that of the small-scale sample and the simulated XRD pattern. Does this phenomenon indicate the large-scale synthesized NTU-98 contain some impurities?

Answer:

We appreciate your suggestion. Due to the addition of a new figure, the original Figure S77 has been renumbered as Supplementary Fig.79.

We fully agree that there are some notable differences between the PXRD pattern of the large-scale synthesized NTU-98 and that of the small-scale synthesized sample. In particular, the diffraction peak width is significantly broader and some peaks exhibit “shoulders” on the low angle side for the large-scale synthesized sample. We attribute these features to the small average size of the crystallites and a broad distribution of the crystallite size of the bulk sample. The change in crystallite morphology is fully expected for the modified synthesis conditions required for scale-up. Although we optimized the reaction parameters, the vigorous stirring employed in our rapid synthesis method may introduce slight heterogeneity in particle size distribution and lead to low crystallinity during the rapid crystal growth process, thereby contributing to the observed peak broadening.

Importantly, our comprehensive characterization demonstrates that these subtle structural variations have negligible impact on the material's performance metrics. Both the gas uptake capacity (Supplementary Fig.79b) and breakthrough separation efficiency (Fig. 4e) remain consistently excellent, confirming that the fundamental adsorption properties and separation capabilities are preserved in the scaled material.

Supplementary Fig.79b. The comparison of C_2H_6 adsorption isotherms for small-scale synthesized and large-scale synthesized.

7. In the introduction, the authors mentioned this pore-environment engineering strategy enhances the framework's hydrophobicity and stability. No information on hydrophobicity was shown in the current version of manuscript. H_2O isotherms at various relative humidity or water contact angles should be added to illustrate the hydrophobicity of NTU-98.

Answer:

Thanks for your suggestion.

We sincerely apologize for the misunderstanding caused by the imprecise wording in our original manuscript. Our intended message was that the hydrophobic methyl groups contribute to the enhanced structural stability of the MOF. Accordingly, we have revised the relevant sections as follows:

“enhancing the water-stability of the framework through steric protection of the metal nodes by hydrophobic methyl groups.”

8. In Figure 4a-4b, the breakthrough curves of the binary gases indicate C_2H_6 and C_2H_4 elutes out earlier than C_2H_2 , which is contradictory with the result from isotherms and ternary breakthrough experiment. Please check.

Answer:

Thank you for pointing this out.

We sincerely apologize for this serious mistake, which was a clear case of mislabeling. This very important issue has now been corrected.

Updated Figure 4a-4b

9. In Page 9, Line 156, the sentence “However, these values yet still lower than that of Zn-FBA (42.8 and 39.8 kJ mol^{-1})” lacks a missing predicate verb (e.g., 'are').

Answer:

Thank you. The sentence was revised as follows:

“However, these values are still lower than that of Zn-FBA (42.8 and 39.8 kJ mol^{-1}).”

10. In Page 11, Line 189, “simpliticty” should be “simplicity”.

Answer:

We apologize for this typo. We have corrected such errors and thoroughly rechecked the entire text.

11. Some formatting issues exist in the figures. For example, the y-axis title in Figure 2d-2e is written as “ C_2H_4/C_2H_2 Uptake difference”, while the labels in Figures 3b and 3c use “Cage Corner N site” and “Cage Corner O site”. These should follow consistent capitalization rules (either title case or sentence case) throughout.

Answer:

Thanks. We have revised Figures 2, 3 and Table 2 to ensure consistent capitalization in all labels. At last, we appreciate your all nice and important comments, which help us to improve the quality of this work a lot.